# Genetic control of DNA methylation is largely shared across European and East Asian populations

Alesha A. Hatton[1], Fei-Fei Cheng[1,2], Tian Lin[1], Ren-Juan Shen[3,4], Jie Chen[4], Zhili Zheng[1], Jia Qu[4], Fan Lyu[4], Sarah E. Harris[5], Simon R. Cox[5], Zi-Bing Jin[3,4], Nicholas G. Martin[6], Dongsheng Fan[7], Grant W. Montgomery[1], Jian Yang[2,8], Naomi R. Wray[1,9], Riccardo E. Marioni[10], Peter M. Visscher[1] & Allan F. McRae[1] ✉

DNA methylation is an ideal trait to study the extent of the shared genetic control across ancestries, effectively providing hundreds of thousands of model molecular traits with large QTL effect sizes. We investigate *cis* DNAm QTLs in three European (n = 3701) and two East Asian (n = 2099) cohorts to quantify the similarities and differences in the genetic architecture across populations. We observe 80,394 associated mQTLs (62.2% of DNAm probes with significant mQTL) to be significant in both ancestries, while 28,925 mQTLs (22.4%) are identified in only a single ancestry. mQTL effect sizes are highly conserved across populations, with differences in mQTL discovery likely due to differences in allele frequency of associated variants and differing linkage disequilibrium between causal variants and assayed SNPs. This study highlights the overall similarity of genetic control across ancestries and the value of ancestral diversity in increasing the power to detect associations and enhancing fine mapping resolution.

Perhaps one of the most notable failures in the study of complex trait genetics is the insufficient representation of ancestral diversity in study participants[1–4]. As such, there is increasing interest in the extent to which the genetic basis of complex human traits is shared across different ancestries[5–7]. For example, Hou et al. utilised admixed individuals to examine the correlation of causal genetic effects across ancestries, finding minimal heterogeneity[8]. Such studies are of particular interest as the extent of shared genetic control across ancestries and the similarity in effect size of associated genetic variants has implications both for the portability of polygenic scores (PGS) and for

disease gene discovery[9,10]. Molecular traits serve as a high throughput way of examining causal genetic effects across ancestries. For example, genome-wide DNA methylation (DNAm) provides hundreds of thousands of molecular phenotypes with large genetic effects[11,12], the combination of which enables substantial insight into the genetic control of DNAm to be obtained with relatively modest sample sizes. DNAm can therefore be used as a model trait for understanding the genetic control of complex traits and disease. In addition, as differential regulation of the genome underlies the majority of the variation in complex traits[13–16], DNAm may elucidate the biological mechanisms

[1]Institute for Molecular Bioscience, The University of Queensland, Brisbane, QLD 4072, Australia. [2]School of Life Sciences, Westlake University, Hangzhou 310030 Zhejiang, China. [3]Beijing Institute of Ophthalmology, Beijing Tongren Hospital, Capital Medical University, 100008 Beijing, China. [4]School of Ophthalmology & Optometry, Wenzhou Medical University, Wenzhou 325027, China. [5]Lothian Birth Cohorts, Department of Psychology, University of Edinburgh, Edinburgh EH8 9JZ, UK. [6]Queensland Institute of Medical Research Berghofer, Brisbane, QLD 4006, Australia. [7]Department of Neurology, Peking University Third Hospital, 100191 Beijing, China. [8]Westlake Laboratory of Life Sciences and Biomedicine, Hangzhou 310024 Zhejiang, China. [9]Queensland Brain Institute, The University of Queensland, Brisbane, QLD 4072, Australia. [10]Centre for Genomic and Experimental Medicine, Institute of Genetics and Cancer, University of Edinburgh, Edinburgh EH4 2XU, UK. ✉e-mail: a.mcrae@uq.edu.au

underlying phenotypic variation due to its potential role in mediating SNP-trait effects. Cumulatively, these factors implicate the ideal role of DNAm for studying the degree of shared genetic architecture of complex traits across ancestries.

Many associations across the genome between DNAm levels and common genetic variants have been identified through DNAm quantitative trait loci (mQTL) analyses[11,17,18]. In fact, mQTLs have been identified at up to 45% of measured locations[19–22], with a large proportion of the genetic control of DNAm located *cis* to the DNAm probe[18,23]. As is seen in complex trait GWAS[24], the majority of studies into the genetic control of DNAm are performed in cohorts of European (EUR) ancestry[11,12,17,18,22,25,26]. Relatively few studies have been performed in other populations[27–31], most of which are limited by sample size. This is despite evidence for population differences in variation of DNAm[28,32–35], with genetic ancestry found to explain the majority of such variation between ancestral groups[28,34]. As such, the degree to which associated genetic variants are shared between populations is not fully understood.

To that end, we performed *cis*-mQTL analysis comparing discovery between EUR and East Asian (EAS) populations using a unified study protocol. This allowed us to assess the degree of shared genetic control between these populations and elucidate differences in genetic architecture driving ancestry-specific associations. We additionally utilise DNAm to investigate the relative improvement in fine mapping resolution of causal variants between single and cross-ancestry fine mapping on a large-scale and to interrogate the presence of ancestry-specific pleiotropic associations with DNAm.

## Results

### Identification of *cis*-mQTLs
We mapped the genetic influences on DNAm in three EUR cohorts and two EAS cohorts with total sample sizes of 3701 and 2099, respectively (Fig. 1 and Table S1). We performed *cis*-mQTL analysis for each DNAm probe by regressing against SNPs 1 Mb upstream and downstream of the target DNAm probe. This was performed individually for each cohort using a unified study protocol. A stringent significance threshold of $p < 10^{-10}$ was used, along with a replication threshold of $p < 10^{-6}$. The most significant SNP for each DNAm probe, referred to here as the lead SNP is considered in the subsequent analysis.

A total of 112,390 (27.8%) DNAm probes with at least one associated SNP that satisfied the $p < 10^{-10}$ significance threshold were identified across all cohorts, of which 19,047 (4.7%) were significant in all five cohorts (Table 1). The limited of overlap across the five cohorts, in part, results from limited statistical power due to sample size, as demonstrated by the relationship between sample size and number of mQTL identified (Table 1). We estimated the correlation of mQTL discovery between cohorts by assessing the correlation in SNP effects, taking into account the standard error of the effect size estimate using a method proposed by Qi et al.[36]. Correlations were calculated between the lead SNPs from the discovery cohort and the corresponding SNP effect in each replication cohort. While the correlation of SNP effects was high between all cohorts ($r_b$ ranging from 0.83 to 0.97), pairwise correlation estimates are largest between cohorts of the same ancestry ($r_b > 0.9$ between EUR cohorts and $r_b > 0.94$ between the two EAS cohorts; Figs. 2 and S1).

### Ancestry based mQTL meta-analysis
Cohort level results were meta-analysed within ancestry (Table 1). There were 129,155 (31.9%) DNAm probes with a significant mQTL identified in at least one ancestry, with 80,394 (62.2% of DNAm probe with significant mQTL) significant at $p < 10^{-10}$ in both ancestries. An additional 18,449 DNAm probes had a significant mQTL in at least one ancestry that previously were not significant in any cohort. There was little difference in the median distance between lead SNPs and DNAm probes across ancestries (6.8Kb for EUR mQTLs and 7.5Kb for EAS mQTL; Table 1). As reported previously[17], among *cis*-mQTLs, significance increases as the distance between the genetic variant and DNAm probe decreases, and this was observed to occur similarly across ancestries (Fig. S2). DNAm probes associated with genetic variation were associated with a median of 115 (IQR = 263) SNPs in EUR and 102 (IQR = 214) SNPs in EAS, reflecting LD between nearby variants.

We estimated the concordance in mQTL discovery between ancestries by assessing the correlation in effect size of the lead SNPs from each ancestry and the corresponding SNP in the alternate ancestry. For each ancestry, lead SNPs were ascertained by $p$ value separately as to minimise upwards bias and the correlation in SNP effects estimated. There was a strong correlation between SNP effects in mQTL discovery for each ancestry ($r_{b\_EUR} = 0.85$ (se 0.002) and

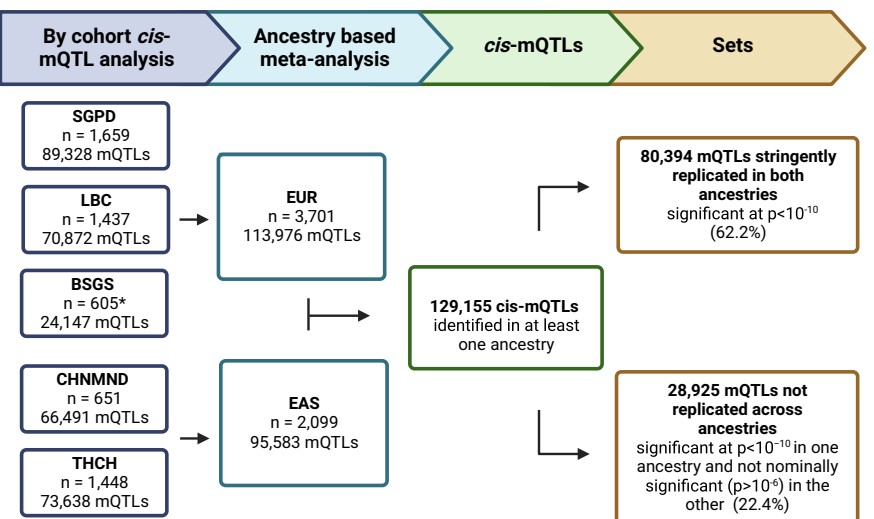

**Fig. 1 | Study overview.** Shown are the number of DNAm probes with significant mQTL detected across the five cohorts and following subsequent ancestry-based meta-analysis. The *cis*-mQTL results were split into two distinct sets to identify those mQTLs stringently replicated in both ancestry and those not replicated across ancestries. A Bonferroni corrected, two-sided $p$ value threshold of $p < 10^{-10}$ was used to define significance at the cohort level from linear regression and mixed linear regression models, and at the ancestry level using inverse variance-weighted meta-analysis. *BSGS cohort includes related individuals from 177 families. Created with BioRender.com.

$r_{b\_EAS}$ = 0.91 (se 0.001); Fig. S3), with the pairwise correlation estimates to be in line with those observed between cohorts of different ancestries.

It has been previously reported in Europeans that DNAm probes with significant mQTL are differentially distributed across genomic features[11]. We investigated whether this was consistent between ancestries using annotations from the Illumina Infinium HM450 manifest file (v1.2). We observed no difference in the genic distribution of mQTLs between ancestries, or in the distribution of CpG island features, illustrating consistency in distributions between ancestries (Fig. S4).

## Shared mQTLs across ancestries

We assessed the degree of shared genetic control of DNAm between EUR and EAS by constructing a set of mQTLs that were stringently significant in both ancestries. This comprised 80,394 DNAm probes where the lead SNP was significant at $p < 10^{-10}$ in both ancestries (62.2% of DNAm probes with significant mQTL). We extracted the lead SNP at each mQTL and the SNP effect in the corresponding ancestry and found very strong agreement in SNP effects between ancestries when accounting for the standard error of the effect size estimate ($r_{b\_EUR}$ = 0.92 (se 0.001), $r_{b\_EAS}$ = 0.94 (se 0.001); Fig. 3a). We compared the DNAm probes that met our criteria for significant mQTLs in both ancestries with the 70,709 DNAm sites where Hawe et al. identified *cis*-mQTLs that replicated in 3799 EUR and 3195 South Asian individuals[31]. Of the 65,522 of the DNAm probes that were in common with our analysis set, 46,332 (70.7% of the DNAm probes) were also significant in both ancestries in our study.

We assessed evidence for putative shared genetic associations for each mQTL that was significant in both ancestries by comparing regional LD (Table 2). In total, 9153 DNAm probes shared the same lead SNP between ancestries (11.4% of the mQTLs significant in both ancestries). For an additional 40,648 (50.6%) the lead SNPs were in strong LD ($r^2$ between lead SNPs >0.8 when assessed in individuals of EUR or EAS ancestry from the 1000 Genomes reference panel (1000G)[37]; Fig. 3b). For those DNAm probes where the lead SNPs were not in strong LD, we quantified the proportion where the lead SNP in each ancestry strongly tagged a common SNP. We constructed LD blocks for each mQTL by identifying SNPs in strong LD ($r^2 > 0.8$) with the lead SNP and compared the overlap in LD blocks between ancestries, identifying 3069 mQTLs with overlapping LD blocks. Thus, a total of 52,870 lead SNP across ancestries are either the same, are in high LD, or strongly tag common SNPs (65.8% of mQTLs that were significant in both ancestries). These observations support the hypothesis of shared

**Table 1 | *Cis*-mQTL analysis identified 112,390 unique DNAm probes with significant mQTL across the five cohorts with within-ancestry meta-analysis increasing discovery mQTLs at 129,155 DNAm probes**

|  | Sample size | Number of DNAm probes with mQTL | Percentage of DNAm probes with mQTL | Median distance between DNAm probe and lead SNP (Q1–Q3; Kb) |
|---|---|---|---|---|
| **Cohort** |  |  |  |  |
| SGPD | 1659 | 89,328 | 22.1% | 6.8 (1.2–23.5) |
| LBC | 1437 | 70,872 | 17.5% | 5.2 (0.7–19.8) |
| BSGS* | 605* | 24,147 | 6.0% | 4.9 (0.7–17.9) |
| CHNMND | 651 | 66,491 | 16.4% | 4.3 (0.7–16.4) |
| THCH | 1448 | 73,638 | 18.2% | 9.1 (2.4–28.1) |
| **Ancestry** |  |  |  |  |
| EUR | 3701 | 113,976 | 28.2% | 6.8 (1.2–24.8) |
| EAS | 2099 | 95,583 | 23.6% | 7.5 (1.7–24.7) |

Shown are the number of DNAm probes with significant mQTL detected in each cohort and each meta-analysed ancestry, alongside the median distance between the DNAm probe and lead SNP. Q1 and Q3 denote the first and third quartiles. A Bonferroni corrected, two-sided *p* value threshold of $p < 10^{-10}$ was used to define significance at the cohort level from linear regression and mixed linear regression models, and at the ancestry level using inverse variance-weighted meta-analysis. *BSGS cohort includes related individuals from 177 families.

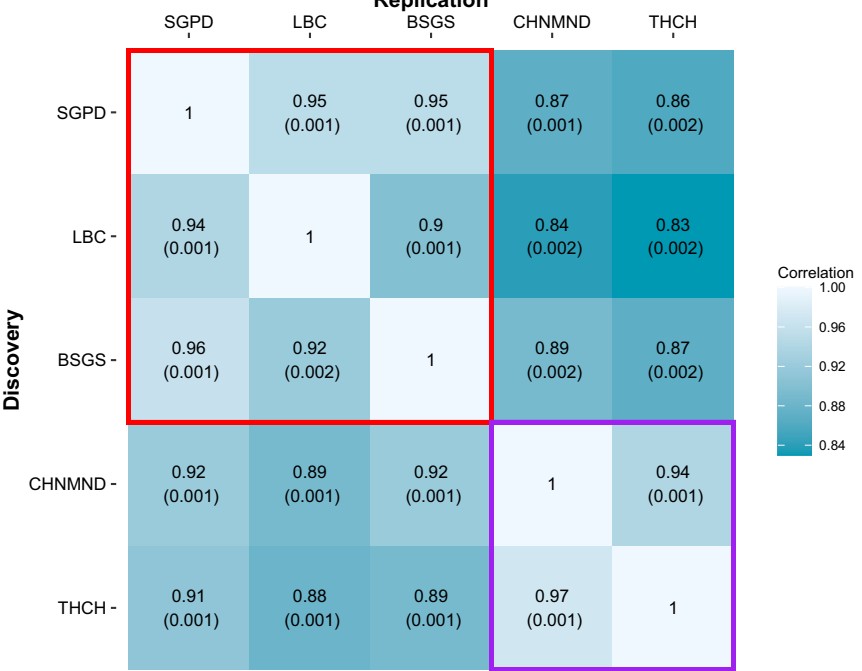

**Fig. 2 | The correlation ($r_b$) of *cis*-mQTL SNP effects across cohorts.** Correlations were calculated between the lead SNP from the discovery cohort and the corresponding SNP effect in the replication cohort. Shown are the estimates of $r_b$ with corresponding standard errors in parentheses. Cohorts of the same ancestry, boxed in red (EUR cohorts) and purple (EAS cohorts), have a stronger correlation of effect sizes than observed across ancestries.

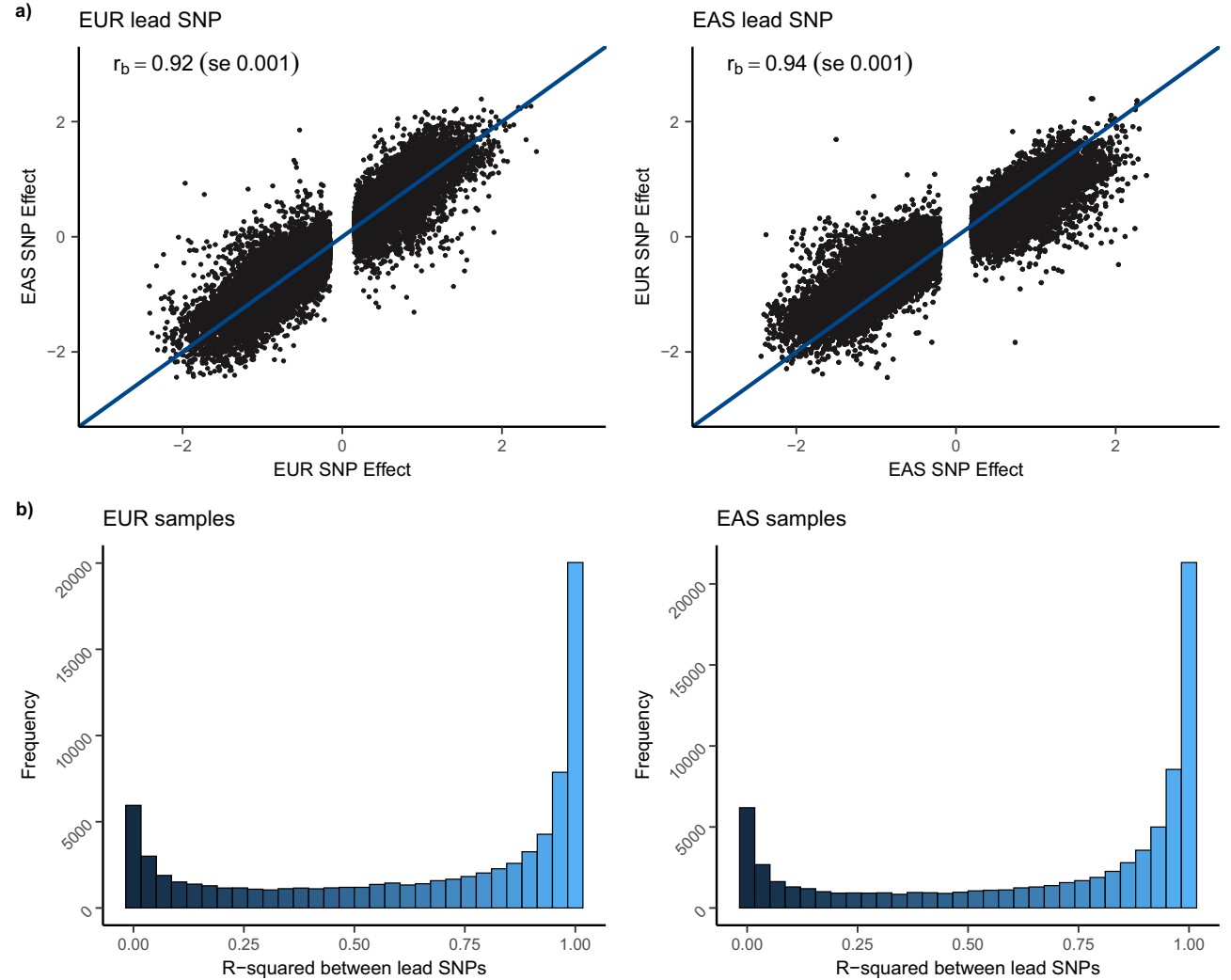

**Fig. 3 | Evidence for shared genetic association at *cis*-mQTLs identified in both populations. a** The correlation ($r_b$) of *cis*-mQTL SNP effects between ancestries for 80,394 DNAm probes with significant mQTL identified in both populations. Shown are effect sizes of the lead SNPs from each ancestry and the corresponding SNP effect in other ancestry. Correlations are presented with corresponding standard errors in parentheses. **b** LD between lead SNPs for DNAm probes with significant mQTL identified in both populations. Shown are the *r*-squared values between lead SNPs calculated using the 1000 Genomes reference panel for the subset of EUR samples (79,391 SNP pairs; left) and EAS individuals (78,108 SNP pairs; right).

**Table 2 | Relationships between EUR and EAS lead SNPs for the 80,394 DNAm probes where the lead SNP was significant in both ancestries**

| Relationship between lead SNPs | Number of mQTLs | Percentage of DNAm probes with mQTL |
|---|---|---|
| Same lead SNP with matching effect direction | 9153 | 11.4% |
| Lead SNPs in strong LD | 40,648 | 50.6% |
| Overlap in LD blocks of lead SNPs | 3069 | 3.8% |
| EAS lead SNP identified as a conditionally independent EUR mQTL | 1031 | 1.3% |
| Overlap in LD block between EAS lead SNP and conditionally independent EUR mQTL | 3999 | 5.0% |
| Total mQTLs with evidence of a putative shared genetic association | 57,900 | 72.0% |

mQTLs which satisfy more stringent (higher) criteria are not carried through to the subsequent (lower) categories. A Bonferroni corrected, two-sided *p* value threshold of $p < 10^{-10}$ was used to define significance from the inverse variance-weighted meta-analysis. Strong LD is defined as $r^2$ between lead SNPs >0.8. LD blocks were constructed for each lead SNP by identifying all other SNPs these were in strong LD with and compared the overlap in LD blocks between lead SNPs for the same DNAm probe. %DNAm probes with mQTL is given with respect to the 80,394 DNAm probes where the lead associated SNP obtained significance at $p < 10^{-10}$ in both ancestries.

signals across ancestries. For the remaining DNAm probes with a significant mQTL in both ancestries, we investigated the presence of multiple independent SNPs within the mQTL window by performing a conditional mQTL analysis. We identified a median of three independently associated SNPs at each DNAm probe in EUR, with the EAS lead SNP identified as a conditionally independent EUR mQTL at the same DNAm probe for 1031 mQTLs (1.3% of mQTLs that were stringently significant in both ancestries). A further 3999 (5.0%) were found to have overlapping LD blocks between the EAS lead SNP and at least one of the conditionally independent EUR mQTLs. Therefore, in

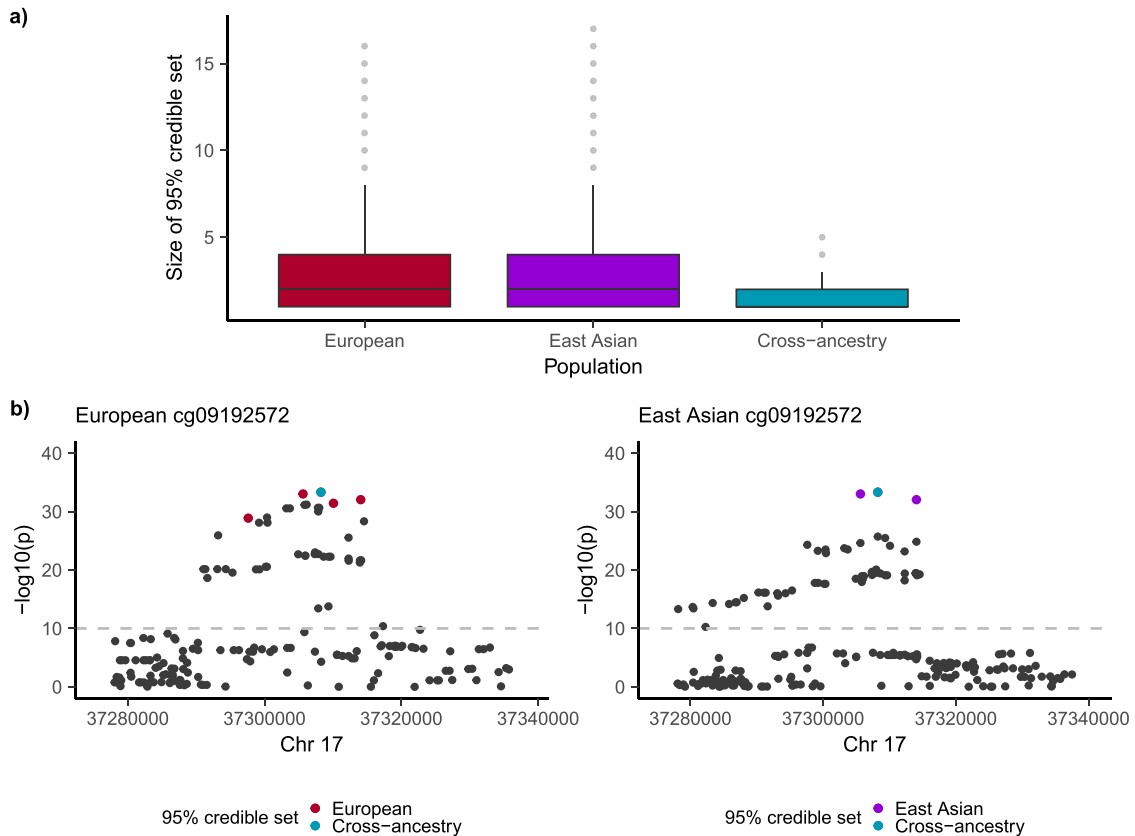

**Fig. 4 | Cross-ancestry fine mapping improves resolution between cohorts of EUR and EAS ancestry. a** Box plot of size of 95% credible set presented for 6385 mQTL associations in EUR, EAS and cross-ancestry populations. Single ancestry fine mapping was performed using SuSiE and cross-ancestry using SuSiEx. The cross-ancestry fine mapping resulted in decreased mean credible set size compared to single ancestry fine mapping performed in samples of EUR and EAS ancestry due to leveraged differences in LD structures across ancestries. Median values are shown in each boxplot, the box denotes the interquartile range and whiskers denote the

rest of the data distribution. **b** Fine mapping of DNAm probe cg09192572 demonstrates improved resolution by incorporating cross-ancestry information. Single ancestry fine mapping using SuSiE generated credible sets of 5 SNPs in EUR shown in red (left) and 3 SNPs in EAS shown in purple (right). The cross-ancestry credible set calculated using SuSiEx resulted in a refined set of 1 SNPs shown in blue. *p* values are from an inverse variance-weighted meta-analysis and are not corrected for multiple testing.

combination there was evidence of a putative shared genetic association at mQTLs for 57,900 of the 80,394 DNAm probes (72.0%) which were significant in both ancestries (Table 2).

## Cross-ancestry fine mapping of shared mQTLs

By leveraging the differences in LD structures across ancestries, we aimed to quantify the gain in resolution for cross-ancestry fine mapping. We did this by contrasting 95% credible set sizes calculated from single and cross-ancestry fine mapping of mQTL summary statistics. For single ancestry fine mapping, to generate credible sets of putative causal variants we employed the Sum of Single Effects (SuSiE) in both EUR and EAS populations[38]. For cross-ancestry fine mapping we used SuSiEx[39] which builds on SuSiE by integrating both EUR and EAS mQTL summary statistics and explicitly modelling population-specific allele frequencies (AF) and LD patterns. For both methods we utilised LD information from 1000G EUR and EAS reference populations for each of the respective ancestries. Fine mapping was performed for mQTLs in the stringent set with the same lead SNP in both ancestries, excluding those in the MHC region. Where multiple mQTLs had the same lead SNP, only those with the smallest *p* value was included leaving 6385 mQTLs. The mean 95% credible set size when fine mapping using SuSiE in a single ancestry was 3.0 and 3.2 for mQTLs identified from EUR and EAS samples respectively (both with median of 2 SNPs). Cross-ancestry fine mapping using SuSiEx yielded a 24.7% and 28.9% reduction in mean credible set size relative to single ancestry

fine mapping in EUR and EAS samples at the same loci respectively (mean cross-ancestry credible set size of 1.5 SNPs, median of 1 SNP; Fig. 4a). As an example, we demonstrate the improvement in fine mapping resolution for DNAm probe cg09192572, where single ancestry fine mapping yielded a credible set of 5 SNPs in EUR and 3 in EAS. In contrast, 1 SNPs was identified in the cross-ancestry credible set (Fig. 4b).

To determine if the improved resolution for the cross-ancestry case was greater than would be obtained from the increased sample size in the EUR ancestry alone, we assessed the expected relationship between increase in sample size and reduction in fine mapping credible set size. We simulated DNAm phenotypes from the above mQTLs using different sample sizes for a subset of EUR samples from the United Kingdom Biobank (UKB) cohort[40]. A single DNAm phenotype was simulated for each mQTL assuming the lead EUR SNP as causal and mQTL analysis performed between the simulated phenotypes and UKB subsets. Fine mapping of the mQTLs using SuSiE was performed using the simulated mQTL summary statistics and the resulting 95% credible set sizes were recorded across DNAm probes and UKB subsets. The simulated mQTLs fine mapped to a similar resolution as the EUR mQTLs for the given sample size. We found a 2-fold increase in the number of a EUR samples resulted in a smaller percentage reduction in credible set size compared to the addition of 2099 EAS samples (Table S2). That is, relative to the baseline of 3701 samples, an increase in sample size of 4099 EUR samples still did not yield the same

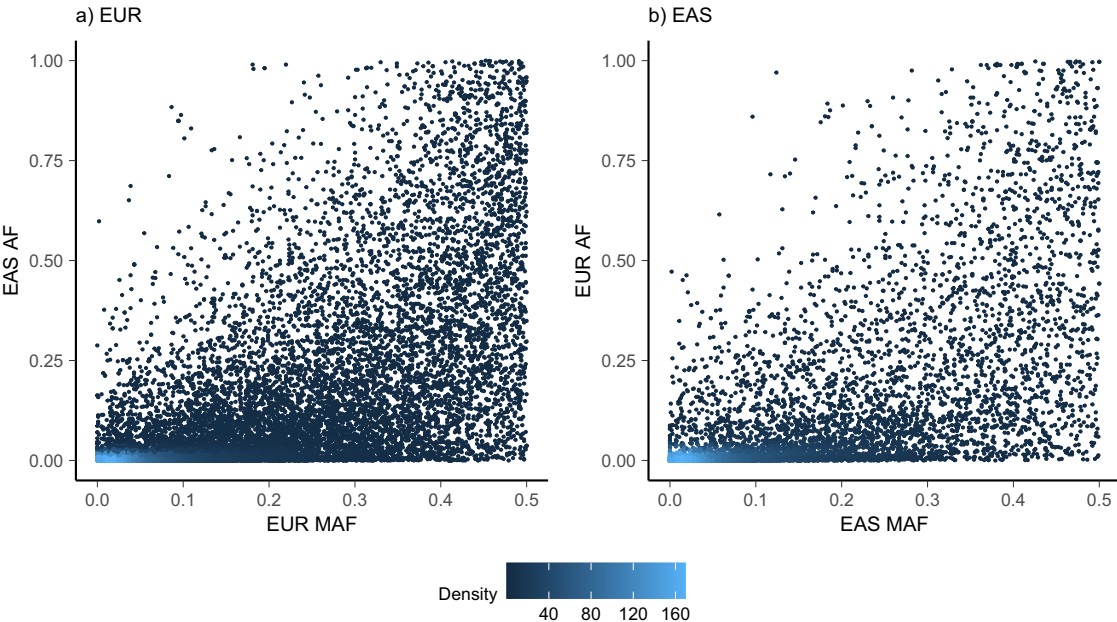

**Fig. 5 | Differences in lead SNP allele frequencies for mQTLs which were significant in a single ancestry only.** Allele frequencies for the EUR lead SNPs for EUR-mQTL (**a**; 21,084 mQTL) and for the EAS lead SNP for EAS-mQTL (**b**; 7841 mQTL). Allele frequencies were calculated using 1000 Genomes reference data for the subset of EUR samples and EAS samples as presented on the respective axes.

percentage reduction in mean credible set size as achieved by the addition of 2099 EAS samples. This demonstrates that across all loci, the improved fine mapping resolution is partly due to cross-ancestry differences in LD structure rather than purely the result of increased sample size.

### Ancestry-specific mQTLs

To identify mQTLs that were ancestry-specific, we constructed a set of DNAm probes where the lead SNP was stringently significant ($p < 10^{-10}$) in one ancestry and was not nominally significant ($p > 10^{-6}$) in the other. Such ancestry-specific mQTLs were classified as either EUR-mQTL or EAS-mQTL based on the significant ancestry. This set encompassed a total 28,925 DNAm probes with ancestry-specific mQTL (22.4% of all DNAm probes with significant mQTL), of which 21,084 were EUR-mQTL (72.9%) and 7841 were EAS-mQTL (27.1%). We excluded mQTLs at 416 DNAm probes which had the same lead SNP in both ancestries as well as 149 mQTLs which were significant in one of the alternate ancestry cohorts as these were potentially due to a lack of statistical power as opposed to being unique to the ancestral group. To determine the proportion of mQTLs driven by only a single cohort within each ancestry, overlap in significance between mQTLs from the ancestry meta-analysis and individual cohorts of the same ancestry was assessed (Table S3 and Fig. S5). Significance was obtained in at least two individual cohorts of the same ancestry for 29.5% of mQTLs (29.8% for EUR- and 28.7% for EAS-mQTLs).

We investigated the sources of heterogeneity underlying the ancestry-specific mQTLs including heterogeneity attributable to differences in the allele frequency and LD of lead variants across ancestries and differences in biological causal effects through examination of effect sizes. We observed considerable heterogeneity when assessing the AF of each lead SNP in the alternate ancestry using 1000G reference panel. For 11,621 (55.1%) EUR- and 3607 (46.0%) EAS-mQTLs, the lead SNP was observed at an extremely rare frequency in the alternate ancestry (minor AF < 0.01; Fig. 5). In comparison, for those mQTLs which were shared across ancestries we observed the lead SNP at an extremely rare frequency in the alternate ancestry for 2.0% and 1.9% of EUR and EAS lead SNPs respectively. We also investigated

the LD between the lead SNP in the ancestry with significant mQTL and the most associated SNP in the region for the alternate ancestry to assess whether there were potential shared mQTLs being missed due to power considerations. Due to lead SNPs being monomorphic in the other ancestry, we were unable to calculate LD between lead SNPs for 4.3% of mQTLs. Of those we were able to calculate, 16,489 (78.2%) of EUR-mQTLs and 6053 (77.2%) of EAS-mQTL were found to have weak or no LD between lead SNPs ($r^2 < 0.2$ using either EUR or EAS 1000G). There were very few mQTLs with strong LD between lead SNPs (3243 with $r^2 > 0.8$ in either EUR or EAS 1000G; 11.2%). We assessed the effect of the lead SNP in the alternate ancestry, noting that 48.0% of lead SNPs were not evaluated in the alternate ancestry during mQTL association testing due to AF filtering in QC. Of those that were evaluated, we observed lack of replication for most mQTLs however some were observed with effects of similar magnitude in the same direction, indicating a small proportion of these "ancestry specific" mQTLs would likely replicate across ancestry with larger sample sizes (Fig. S6). This suggests that while differences in biological causal effects play a role, the observed heterogeneity in mQTLs across ancestries is largely driven by differences in the allele frequencies.

We demonstrate the utility of these ancestry-specific mQTLs by identifying pleiotropic associations with common genetic variation associated with 220 traits from Sakaue et al.[41] (Supplemental Data 1). Using SMR[42], we tested if the SNP effects on the traits were mediated by DNAm at 28,925 ancestry-specific mQTLs. Across 57 of these traits, we identified 163 ancestry-specific associations with DNAm ($p_{SMR} < 10^{-6}$), of which 147 were unique in EUR and 16 in EAS, with non-significant heterogeneity ($p_{HEIDI} > 0.01$; Supplementary Data 2 and Supplementary Fig. 7). We note that a potential driver of these difference would be low AF in the alternate ancestry which is likely the case for 115 (70.6%) of the associations (108 identified in EUR and 7 in EAS), however the remaining 48 pleiotropic associations occur at SNPs with common AF in both ancestries (minor AF > 0.05). For example, we identified a unique pleiotropic association in individuals of EUR ancestry with probe cg19197236 in chromosome 2 with Glucocorticoid use (R03BA) (Fig. 6), with no association for either DNAm or R03BA in

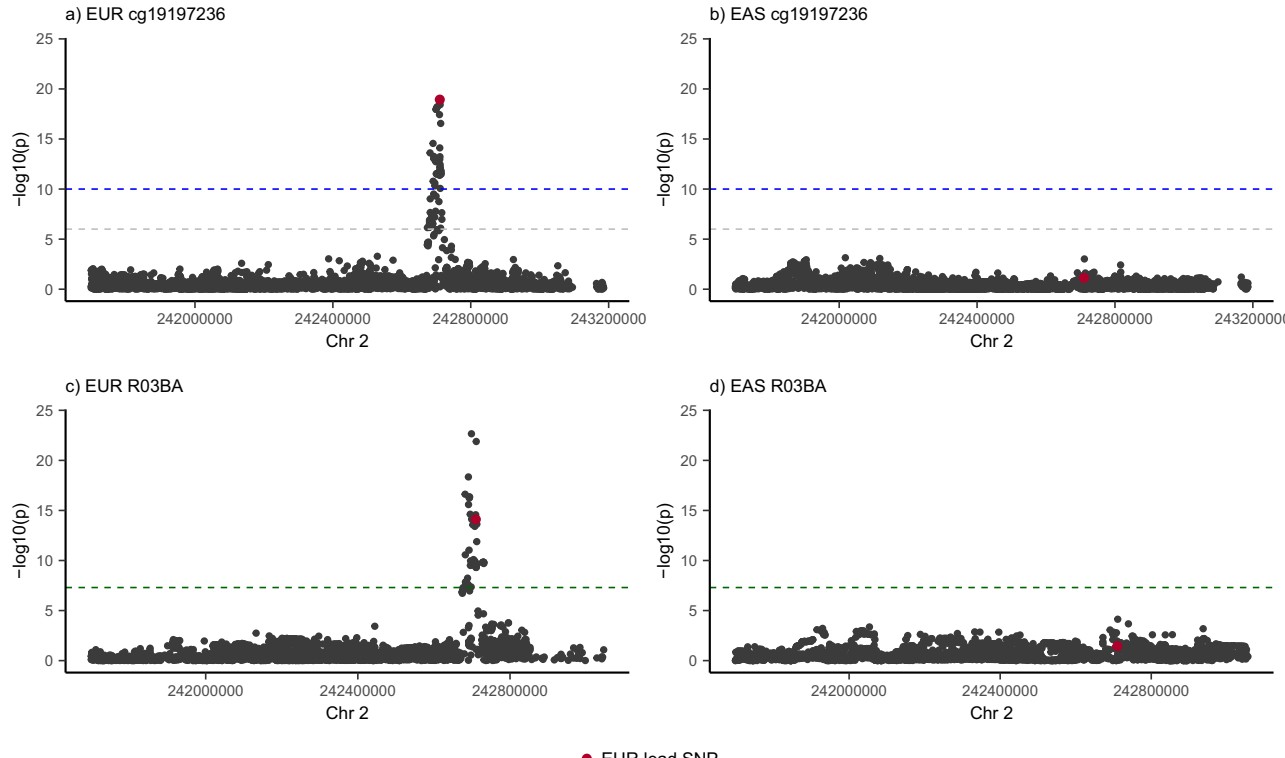

**Fig. 6 | Evidence of ancestry-specific pleiotropic associations.** mQTL and GWAS association for Glucocorticoid use (R03BA) in populations of EUR and EAS ancestry. Shown is genomic region surrounding DNAm probe cg19197236 on chromosome 2 identified in mQTL analysis and R03BA GWAS. The presence of an associated genetic signal was observed in EUR with both DNAm (**a**) and R03BA (**c**), with the pleiotropic nature of the mQTL confirmed using SMR. Associations were not observed for either DNAm or R03BA in EAS (**b**, **d**, respectively). mQTL and GWAS $p$ values were from linear regression analyses. The blue line indicates the significance threshold ($p < 10^{-10}$) in mQTL analysis, the grey line is the replication threshold ($p < 10^{-6}$) and the green line indicated the significance threshold ($p < 5 \times 10^{-8}$) in GWAS. The SNP highlighted in red is the EAS lead SNP used as the exposure outcome in SMR analysis.

individuals of EAS ancestry (minor AF of SNP rs7608524 of 0.37 in EUR and 0.41 in EAS).

## Discussion

Whilst multiple studies have previously examined the genetic influences of DNAm, there is limited literature comparing mQTLs between populations of different ancestries[27,28,30,31]. We present a *cis*-mQTL study comparing discovery between EUR and EAS populations using a unified study protocol to quantify empirically the degree of shared and heterogenous genetic control between ancestries. Within ancestry meta-analysis increased discovery to 129,155 mQTLs (31.9% of the 404,503 DNAm probes analysed) identified in at least one ancestry, an increase of 18,449 (4.6%) which were previously not significant in any cohort.

Our study is primarily focused on the identification and characterisation of mQTLs which were shared across ancestries and those which were unique to a single ancestry to elucidate differences in the genetic architecture of associations across populations. We observed 80,394 associated mQTLs (62.2% of DNAm probes with significant mQTL) to be significant in both ancestries, pointing to a shared genetic basis of DNAm, while 28,925 mQTLs (22.4%) were unique to a single ancestry. Discovery of shared mQTLs was compared with Hawe et al.[31] which identified *cis*-mQTLs at 70,709 DNAm sites that replicated in 3799 EUR and 3195 South Asian individuals and of the 65,522 of the DNAm probes which were in common with our analysis set, 46,332 (70.7% of the DNAm probes) were similarly stringently significant in both ancestries. Our study demonstrated mQTLs effect sizes to be highly conserved across populations, with large, positive correlations observed between all cohorts, indicating the genetic underpinning of DNAm operates consistently across EUR and EAS ancestries. Despite

this, pairwise correlation estimates were largest between cohorts of the same ancestry. This is consistent with results observed from Kassam et al., which found correlation estimates of *cis*-mQTLs SNP effects to be larger in samples that are more genetically similar, and smaller for those of ancestries that are more genetically distant when contrasting populations of Indian, Chinese, Malay, Bangladesh and EUR ancestry[30].

We show that differences in mQTL discovery are likely due to differences in AF of associated variants between populations and differing LD between causal variants and assayed SNPs across populations. For mQTLs that were identified in only a single population, the discovery was skewed toward associated variants that had lower AF in the other population. As a consequence, we noted stark differences in the LD spectrum between unique mQTL and those mQTLs shared between ancestries, with a lack of LD between lead SNPs for 77.9% of ancestry-specific mQTL. Such evidence of heterogeneity in associations between the two ancestries likely results from both population-specific variants and shared variants that were undetected in the second population due to AF and LD profiles. Further, this study allowed us to demonstrate the utility of ancestry-specific mQTLs by evaluating the presence of associations similarly identified across 220 traits and subsequent pleiotropic relationships.

We note that an undetermined proportion of the absence of replication between ancestries may be attributed to study power. We sought to quantify this by comparing mQTLs identified in EAS and not replicated in EUR with the largest mQTL study to date (GoDMC), a meta-analysis of 27,750 EUR participants[22]. Of the 4903 EAS-mQTLs that were in common, 4032 (82.2%) were similarly significant in GoDMC using their significance threshold of $p < 10^{-8}$. For comparison, of the 97,510 mQTL probes significant in our EUR discovery that were

in common with GoDMC, 97,382 (99.9%) were similarly significant in GoDMC (regardless of replication in EAS). This suggests that our identification of ancestry-specific mQTLs is limited by power although despite this there indeed appear to be some mQTLs which are unique and are not replicated in EUR samples even in the presence of large sample sizes.

The utility of cross-ancestry fine-mapping in reducing the number of putative causal variants has been previously demonstrated for many complex traits[2,7,43–47] including schizophrenia[46], type 2 diabetes[48,49], BMI[50], RA[51] and height[7]. The success of such methods is supported by evidence that that association signals are driven by variants shared across populations[2,7,52], despite observed heterogeneity in effect sies[7,46,52]. By utilising DNAm as a model complex trait, we identified a mean reduction in putative causal set size of 24.7% between EUR and cross-ancestry fine mapping across all loci. By simulating mQTLs from the UKB using differing sample sizes, we demonstrated more than a 2-fold increase in the number of EUR samples would be required to achieve the same percentage reduction in causal set relative to the addition of 2099 EAS samples. This suggests that the improved fine-mapping resolution across these mQTL loci is more likely the result of differences in LD structure between populations than the result of increased sample size alone.

Whilst there is evidence of many mQTL in *trans*[11,18,21,26], we did not consider *trans* regions due to their much smaller effects that would amplify issues of power when comparing across ancestries. Additionally, variation in DNAm has been associated with multiple environmental exposures such as smoking[53], sun exposure[54] and exercise[55,56], with some also sharing a genetic component[57,58]. The relationship between DNAm, genetic variation and environmental exposures may explain some of the ancestral heterogeneity observed which could be further explored using gene–environment (G × E) analysis. Lastly, we note that genetically determined ancestry may not reflect an individual's self-reported ethnicity, and that we are examining a limited spectrum of the ancestral diversity.

In summary, mapping the genetic factors associated with DNAm allowed us to gain insight into the large degree of shared genetic control between EUR and EAS populations and enhanced our understanding of the underlying differences in genetic architecture driving ancestry-specific associations. This study highlights the importance of incorporating ancestral diversity into genomic studies, both to identify causal variants through increased power to detect associations as well enhanced fine mapping resolution.

## Methods
### Study cohorts
Study cohorts are summarised in Table S1. All study cohorts are of either EUR or EAS genetic ancestry and contain DNAm derived from whole blood samples.

**Systems Genomics of Parkinson's Disease (SGPD).** The SGPD cohort comprises genotype, phenotype and DNAm data for 2333 participants (1292 PD cases and 1041 controls)[59]. Participants were recruited as part of three different studies across Australia and New Zealand. Controls consist of healthy community-based, age-matched volunteers residing in the same area and from the same ethnic background as the PD patients, together with patients' spouses and siblings. DNAm and genotyping was performed as reported in Vallerga et al.[59]. The SGPD DNAm data are available from the Gene Expression Omnibus (GEO) under accession code GSE145361.

**Lothian Birth Cohorts of 1921 and 1936 (LBC).** The LBCs are longitudinal studies of aging focusing on lifetime cognitive change[60,61], containing biomarker, genetic, DNAm and cognitive phenotype data (https://www.ed.ac.uk/lothian-birth-cohorts). The LBC1921 and 1936 cohorts are follow-up studies of the Scottish Mental Health Surveys of 1932 and 1947, with subjects at an average age of 79 years and 70 years, respectively. An overview of the data collected can be found in the cohorts' profile article[60,62] and has been described extensively elsewhere[12,63]. In the analysis we included the first wave of LBC data. The LBC DNAm data is available at the European Genome-phenome Archive under accession number EGAS00001000910.

**Brisbane Systems Genomics Study (BSGS).** The BSGS is a family-based study, consisting of adolescent monozygotic and dizygotic twins, their siblings and parents[12]. DNAm was measured on 614 individuals from 177 families of EUR descent. Children have a mean age of 14 years (range 9–23) and parents 47 year (range 33–75). A full description of the BSGS cohort has been previously provided[12,63,64]. DNAm data for the BSGS is available from the Gene Expression Omnibus under accession code GSE56105.

**Chinese Motor Neuron Disease Cohort (CHNMND).** The CHNMND is an Amyotrophic Lateral Sclerosis (ALS) case-control cohort. Participant recruitment has been described previously[65,66]. DNAm data was available for 453 cases and 198 controls, excluding those with familial Motor Neuron Disease (MND). Deposit of DNAm data in a repository does not comply with the consent process and ethics approval, but sharing data is possible by emailing the corresponding author of the above cohort publications.

**Tibetan-Han Chinese high-altitude (THCH).** The THCH cohort is a study on high-altitude adaptation comprised of three groups of EAS ancestry: 918 NHs (highland Tibetan Chinese), 348 AN (highland Han Chinese), and 488 NLs (lowland Han Chinese). Tibetan Chinese subjects were recruited from two sites in the Tibetan Plateau (TP), both at ~4100 m above sea level, while Han Chinese subjects were collected from TP and Wenzhou (Wz), as previously described[67]. Primary data of the Tibetan and Han Chinese subjects are available through application at https://www.wmubiobank.org. The median age across all samples was 38 years old (ranging from 11 to 90) with 74.6% of sample being female. After standard quality control (QC), 1448 unrelated individuals were retained.

### Genotype imputation and quality control (QC)
Genotyping methods for each cohort are displayed in Table S1. SNP genotype QC was performed individually for each cohort following standard protocols[68]. Genotype data were restricted to autosomes and SNPs were excluded for missing genotype call rate (>5%), marked departure from Hardy–Weinberg equilibrium (HWE; $p < 10^{-6}$) and low minor allele frequency (AF < 1%) using PLINK (v2.0)[69]. In order to classify cohorts into genetically similar groups we inferred ancestry based on principle component analysis using genetic data. We note that genetically determined ancestry may not reflect an individual's self-reported ethnicity and that we are examining a limited spectrum of the ancestral diversity. Using the GCTA software (v1.93.2beta)[70], principal components were projected against the 1000 Genomes reference panel (1000G)[37]. A threshold of six standard deviations from the mean of each population cluster was used to remove outlying individuals. SNPs were excluded based on deviance in minor AF from 1000G by comparing minor AF between each cohort and the ancestry-matched reference panel. Observations deviant from the threshold curve of 3 times the standard deviation were determined to be outliers. For the THCH cohort SNPs identified from a GWAS between TP Han and Wz Han subjects were excluded to limit confounding from population stratification. Further QC exclusionary measures for subjects were: missing genotype call rate (>5%); suspected sample error or contamination indicated by high heterozygosity or indeterminate genetic sex.

For each cohort, SNPs were phased using EAGLE2 (v2.0.5) + PBWT[71] and imputed against 1000G Phase 3 reference panel

using Sanger Imputation Service[72]. The 1000G imputation panel was specifically selected over larger reference panels such as the Haplotype Reference Consortium, as it was collected from 26 populations in AFR, EAS, Europe, South Asia, and the Americas. This increase in coverage of non-EUR individuals increase the likelihood of even imputation accuracy across ancestries. A GRM was calculated using GCTA to determine the relatedness between individuals based on genotyped SNPs. Related individuals were removed using a 5% threshold or controlled for in downstream analysis for familial data. Imputed SNPs were then filtered based on info score (<0.8), HWE ($p < 10^{-6}$) and minor AF (<1%). Only samples with available DNAm data were brought forward for analysis with the number of SNPs, post imputation and QC, are provided in Table S1.

## DNAm QC and normalisation

For each of the cohorts, all individuals had DNAm assayed on the Illumina Infinium HumanMethylation450 BeadChip. DNAm levels for each probe were obtained as the ratio of the methylated probe intensity to the overall intensity and expressed as Beta-values ranging between 0 and 100%, representing the proportion of cells methylated at each site. Normalisation and QC of the DNAm data were performed independently for each cohort using the R package meffil[73]. Standard QC threshold parameters were used to remove individuals with outlying methylated/un-methylated ratios, high missing calls, discordance between reported and estimated sex, and probes with excess missingness. Predicted cell counts (Bcell, CD4T, CD8T, Mono, Neu and NK) were calculated using the Houseman algorithm implemented in meffil[74]. Probes on the sex chromosomes or those annotated as binding to multiple sites across the genome were excluded[75]. After cleaning, 404,503 DNAm probes which were common across all cohorts remained for association analysis.

## DNAm adjustment

Within each cohort, individual probes were pre-corrected using a linear mixed model (R package lme4[76]), adjusted for sex, age, age² and estimated cell proportions to account for differences in DNAm between samples. Random variation was accounted for from array slide and row effects. Cohort effects were also pre-corrected including case-control status for SGPD and CHNMND cohorts, and sample origin (Han/Tibetan) and altitude for THCH cohort. A rank based inverse normal transformation was performed on the residuals to counteract departures from normality to reduce the false positive rate.

## Cis-mQTL analysis

An overview of the *cis*-mQTL analysis pipeline is provided in Fig. 1. To map genetic influences on DNAm, we performed *cis*-mQTL analysis for each DNAm probe by regressing against SNPs within a 1 Mb window on either side of the target DNAm probe. Analysis was restricted to measured probes which passed QC in all five cohorts. Association tests for the LBC, SGPD and CHNMND cohorts were performed using –glm in PLINK2.0[77], while –eqtl in OSCA[78] was used to analyse the THCH cohort, with the difference in methodology due to external data custodianship. The BSGS cohort was analysed using the fastGWA in GCTA[70,79] which controls for familial relatedness using a sparse GRM with a threshold of 0.05[80]. This has been shown to capture nearly identical proportions of phenotypic variance as the full GRM. A stringent $p = 0.05$ Bonferroni correction of $p < 10^{-10}$ was applied to account for multiple testing based on the number of probes analysed and the approximate number of independent SNPs in the 2 Mb window (1000 SNPs per 1 Mb). The replication threshold was set at $p < 10^{-6}$ which is Bonferroni corrected for the approximate number of independent mQTLs. The most significant SNP for each DNAm probe was retained and is referred to here as the lead SNP.

## Within-ancestry meta-analysis

We performed meta-analysis (fixed effects, standard inverse-variance weighted) to combine mQTL associations from cohorts of the same ancestry (see Table S1). This was performed using –meta in OSCA (v0.46)[78].

The mQTL results were split into two distinct sets: Stringent set—DNAm probes where the lead SNP was stringently significant in both EUR and EAS ancestries ($p < 10^{-10}$). This set aimed to determine mQTL that were shared across ancestries; Not replicated set—DNAm probes where the lead SNP was stringently significant in one ancestry ($p < 10^{-10}$) and was not nominally significant in the other ($p > 10^{-6}$). This set aimed to determine mQTL which were unique to a single ancestry (while acknowledging differences in power). For ease of explanation, DNAm probes with mQTL that were stringently significant in one ancestry and not replicated in the second are denoted by the ancestry with the stringently significant mQTL (EUR-mQTL or EAS-mQTL).

## LD patterns for *cis*-mQTLs between ancestries

The pairwise-correlation between alleles of the lead SNP for mQTLs in each ancestry was calculated to determine if mQTLs were in LD. LD was calculated using 1000G as an external reference panel, individually for the subset of 503 EUR (EUR-1000G) and 504 EAS individuals (EAS-1000G). In addition, LD blocks were constructed for lead SNPs using the corresponding ancestry-specific subset of 1000G, with LD blocks defined as SNPs with $r^2 > 0.8$.

## Conditional analysis

To identify DNAm probes with multiple independent genetic associations, we conducted a conditional and joint analysis (COJO)[81] using GCTA[70]. We accounted for the correlation structure between SNPs within a 10 Mb window using Health and Retirement Study (HRS; $n = 8652$)[82] imputed to 1000G as an external LD reference panel. This was performed for each mQTL using meta-analysed EUR summary statistics. Similar analysis was not performed using EAS meta-analysed summary statistics due to the lack of appropriately large reference panel (recommended minimum sample size of 4000). We retained SNPs that had conditional $p < 10^{-10}$.

## Fine mapping

We aimed to quantify the gain in fine mapping resolution that results from leveraging differences in LD structures across ancestries. This was performed by contrasting credible set sizes calculated from single and cross-ancestry fine mapping of mQTL summary statistics. For single ancestry fine mapping, to generate credible sets of putative causal variants we employed the Sum of Single Effects (SuSiE)[38] in each ancestry. This model uses a Bayesian modification of simple forward selection to generate credible sets which are designed to have high probability to contain a variable with non-zero effect, while at the same time being as small as possible. For cross-ancestry fine mapping we used SuSiEx[39] which builds on SuSiE by integrating both EUR and EAS mQTL summary statistics and explicitly models population-specific AF and LD patterns. For both methods we utilised LD information from 1000G for the respective ancestries and set the number of putative causal signals to be one, with credible sets comprised of SNPs with a cumulative 95% posterior probability of being causal. Fine mapping was performed for each of the mQTLs in the stringent set with the same lead SNP in both ancestries. Where multiple mQTLs had the same lead SNP, only the mQTL with the smallest $p$ value was included and those mQTLs in the MHC region were excluded, resulting in 7416 mQTLs for fine mapping. mQTLs were excluded where the putative number of causal SNPs, in at least one ancestry, was more than three IQR from the mean, resulting in 6385 mQTLs remaining.

## Fine-mapping simulation

Simulations were performed to determine the expected relationship between the increase in sample size and the reduction in fine mapping credible set size. The above mQTLs were brought forward for simulation. Data from the UK Biobank (UKB) was utilised for simulations. The full release of the UKB data consisted of genotype and phenotype data for ~500,000 participants across the United Kingdom[40]. A subset of individuals of European ancestry ($n = 456,422$) was identified by projecting the UKB PCs onto those of the 1000 Genome Project with related individuals removed using GCTA (GRM > 0.05). Subsets of the UKB were selected to match the EUR and cross-ancestry sample sizes ($n = 3701$ and 5800) with subsequent increases of 1000 sample increments ($n = 6800, 7800, 8800, 9800$). A single DNAm phenotype was simulated for each mQTL assuming the lead EUR SNP as causal using GCTA, using the EUR lead SNP effect estimate. Simulated DNAm levels were rank normalised and mQTL analysis performed between the simulated phenotypes and UKB subsets using PLINK2.0. The resulting *cis*-mQTL summary statistics were fine-mapped using SuSiE and 95% credible set sizes were recorded across DNAm probes and UKB subsets, filtering outliers using the same criteria as above resulting in putative credible sets for 4528 simulated mQTLs.

## Summary-based Mendelian randomisation (SMR)

To demonstrate the utility of ancestry-specific mQTLs, we investigated the presence of pleiotropic associations between DNAm and 220 traits as reported in Sakaue et al.[41]. Summary statistics of analysis performed in individuals of EUR ancestry were obtained from meta-analyses with the UK Biobank and FinnGen while EAS was from BioBank Japan (Supplementary Data 1). We utilised the SMR software (v1.03) to test if the effect size of a SNP on the phenotype is mediated by DNAm that can subsequently be used to prioritise genes underlying GWAS hits for follow-up functional studies[42]. A key assumption of this approach is that the same set of underlying causal variants determines both DNAm and the trait. We utilised 28,925 ancestry-specific mQTLs, of which 21,084 were EUR-mQTLs and 7841 EAS-mQTLs and applied these to the 220 traits. To ensure the pleiotropic signals were ancestry-specific, mQTLs were excluded where a GWAS signal was present within the *cis*-window in the alternate ancestry. SMR significant results were declared at $p_{SMR} < 10^{-7}$, which is Bonferroni corrected for the approximate number of DNAm probes tested. While significant SMR test results implicate a role for the mQTL, SNPs passing the SMR heterogeneity in dependent instruments (HEIDI) test ($p_{HEIDI} > 0.01$) have robust support for the direct causal or pleiotropic relationships of the trait-associated SNPs influencing DNAm.

## Reporting summary

Further information on research design is available in the Nature Portfolio Reporting Summary linked to this article.

## Data availability

The mQTL summary data from the meta-analysis of samples of each European ($n = 3071$) and East Asian ($n = 2099$) ancestry generated in this study are available at https://yanglab.westlake.edu.cn/software/smr/#mQTLsummarydata. These results have been provided in SMR BESD format (see https://yanglab.westlake.edu.cn/software/smr/#BESDformat). Access to individual level data for each of the cohorts is as follows: The SGPD DNAm data are available from the Gene Expression Omnibus (GEO) under accession code GSE145361. The LBC DNAm data are available at the European Genome-phenome Archive under accession number EGAS00001000910. DNAm data for the BSGS are available from the Gene Expression Omnibus under accession code GSE56105. Deposit of CHNMND DNAm data in a repository does not comply with the consent process and ethics approval, but sharing data is possible by emailing the corresponding author of the cohort publication. Primary data of the Tibetan and Han Chinese subjects are available through application at https://www.wmubiobank.org. Analysis of the UK Biobank resource was conducted under the application number 12505. The genotype and phenotype data are available upon application to the UKB (http://www.ukbiobank.ac.uk/). Health and Retirement Study data were accessed from dbGaP (accessions: phs000428). The web links for the publicly available datasets used in the study are as follows: The 1000 Genomes Phase 3 data available at https://ftp.1000genomes.ebi.ac.uk/vol1/ftp/phase3/; mQTL data from Hawe et al.[31]: https://zenodo.org/record/5196216#.YRZ3TfJxeUk; mQTL data from Min et al.[83] data: http://mqtldb.godmc.org.uk; GWAS summary statistics for 220 traits used for SMR from Sakaue et al.[41]: https://pheweb.jp/downloads; Annotation of Infinium DNA Methylation BeadChip probes: https://zwdzwd.github.io/InfiniumAnnotation.

## Code availability

The publicly available software tools used for data analysis are described in the "Methods" with the web links as follows: plink (v2.0): https://www.cog-genomics.org/plink/2.0; OSCA (v0.46): https://yanglab.westlake.edu.cn/software/osca; GCTA (v1.93.2beta): https://yanglab.westlake.edu.cn/software/gcta/#Overview; susieR: https://stephenslab.github.io/susieR; susieX: https://github.com/getian107/SuSiEx; SMR (v1.03): https://yanglab.westlake.edu.cn/software/smr/.

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

## Acknowledgements

Acknowledgements for cohort data collection and processing are provided in Supplementary Note 2. This work was supported by the Australian National Health and Medical Research Council (1113400 P.M.V., 1173790 N.R.W., 1177194 G.W.M.) and the Australian Research Council (FL180100072 P.M.V., FT200100837 A.F.M.). A.A.H. is supported by an Australian Government Research Training Program (RTP) Scholarship. F.C. is supported by the Dr Jian Zhou Memorial Scholarship.

## Author contributions

Design and conception: A.F.M. and P.M.V. Data collection: R.J.S., J.C., J.Q., F.L., S.E.H., S.R.C., Z.B.J., N.G.M., D.F., G.W.M., J.Y., N.R.W. and R.E.M. Data cleaning and quality control: T.L., A.A.H., F.C. and Z.Z. Data analysis: A.A.H., F.C. and Z.Z. Interpretation of data: A.A.H., A.F.M., P.M.V., J.Y., N.R.W. and R.E.M. Drafting manuscript: A.A.H. and A.F.M. All authors contributed to manuscript review and revision.

## Competing interests

R.E.M. is a scientific advisor to the Epigenetic Clock Development Foundation and Optima Partners. The remaining authors declare no completing interests.
