## [Peer Review File · Nature Communications]

Genetic control of DNA methylation is largely shared across European and East Asian populationsREVIEWER COMMENTS

Reviewer #1 (Remarks to the Author):

The manuscript by Hatton et al presents analyses of DNA methylation in a large cross ancestry data set showcasing two important findings. First, that genetic control of methylation is largely shared and consistent across European and Asian ancestry populations. Secondary, it highlights that genetic control of methylation is consistent across different methylation data sets. Although largely expected these findings are important to report and constitute a nice addition to the field. The manuscript is clearly and concisely written showcasing the main results and its context in the wider field. As such I fully support its publication.

I hope the comments below are useful for the authors to improve their work.

1. I encourage the authors to be more didactic in their presentation of heterogeneity in genetic effects by ancestries. As noted in several parts of the manuscript (i.e. "Ancestry-specific mQTLs"), the authors find most of the heterogeneity is due to differences in frequencies (and LD) across ancestries with little impact from actual differences in biological causal effects (change in methylation per allele substitution). The manuscript could be improved if the authors clarify in text the distinction among multiple types of heterogeneity (allelic effect vs frequency difference).
2. Very high correlation in genetic effects genome-wide does not exclude local heterogeneous effects at given mQTL sites. It would be interesting for the authors to run a standard test for heterogeneity by ancestry(or study) for every mQTL and report sites where such heterogeneity exists.
3. Fig 2 is very interesting showcasing the main discovery of high correlation. Can the authors also investigate the correlation in the phenotypes (i.e. methylations) across data sets/ancestries? Are they of the same magnitude or lower due to different environments?
4. Discussion about the utility of cross-ancestral data could be enhanced; some of the first studies showing how fine-mapping could be drastically improved in cross ancestry data date back to more than a decade ago in studies such as Zaitlen et al AJHG 2010 (PMID: 20085711).

Reviewer #2 (Remarks to the Author):

Authors conducted methylation QTL (mQTL) study meta-analysis of Europeans and Asians. In total, 5 cohorts with ~5700 individuals were included. Cross-population mQTL meta-analysis identified that mQTL loci were generally shared. Ancestry-specific QTL were mostly due to different allele frequency spectra between populations. They found fine-mapping of putative mQTL by cross-population approaches. Some overlap with the GWAS variants were observed. While this is a typical mQTL study, findings and concepts were not novel.

1. It was not clear for the reviewer what was the main findings and novelty of the study. Certain sample size may provide mQTLs, but how the results overwhelmed the previous similar studies were not clear.
2. No novel conceptual findings. Cross-population study can provide statistical power. It is not surprising.
3. Authors concluded that majority of the ancestry-specific mQTL were due to allele frequency differences. It was not exciting. More detailed assessments on truly ancestry-specific mQTL with difference effect sizes across populations were warranted.

4. In cross-populations comparisons, fine-mapping should be applied.
5. Functional annotation of the mQTL variants can be done in depth.
6. As for GWAS risk variant overlap, much wider ranges of GWAS phenotypes should be applied.
7. This study can have a resource value. But no descriptions on data deposit, which should lower the value.

Peer Review File

Manuscript Title: Genetic control of DNA methylation is largely shared across European and East Asian populations

Reviewers' comments and author rebuttals

We thank the two reviewers for their constructive comments and suggestions, which have helped us to improve this manuscript. We have responded to all the reviewers' comments point-by-point below.

Reviewer #1:

The manuscript by Hatton et al presents analyses of DNA methylation in a large cross ancestry data set showcasing two important findings. First, that genetic control of methylation is largely shared and consistent across European and Asian ancestry populations. Secondary, it highlights that genetic control of methylation is consistent across different methylation data sets. Although largely expected these findings are important to report and constitute a nice addition to the field. The manuscript is clearly and concisely written showcasing the main results and its context in the wider field. As such I fully support its publication.

I hope the comments below are useful for the authors to improve their work.

1. I encourage the authors to be more didactic in their presentation of heterogeneity in genetic effects by ancestries. As noted in several parts of the manuscript (i.e. "Ancestry-specific mQTLs"), the authors find most of the heterogeneity is due to differences in frequencies (and LD) across ancestries with little impact from actual differences in biological causal effects (change in methylation per allele substitution). The manuscript could be improved if the authors clarify in text the distinction among multiple types of heterogeneity (allelic effect vs frequency difference).

We thank the reviewer for their valuable comments and have endeavoured to be more didactic in the presentation of ancestry-specific mQTLs. The following text has been added to the section on ancestry-specific mQTLs:

"We investigate the sources of heterogeneity underlying the ancestry-specific mQTLs including heterogeneity attributable to differences in the allele frequency and LD between lead variants identified in each ancestry and differences in biological causal effects through examination of effect sizes.

....

This suggests that while differences in biological causal effects play a role, the observed heterogeneity in mQTLs across ancestries is largely driven by differences in the allele frequencies. "

2. Very high correlation in genetic effects genome-wide does not exclude local heterogeneous effects at given mQTL sites. It would be interesting for the authors to run a

standard test for heterogeneity by ancestry (or study) for every mQTL and report sites where such heterogeneity exists.

While we agree that such an analysis could be useful, we don't know the causal variants and any test for heterogeneity would be confounded by the differences in LD structure between the two populations. Those mQTLs with the same lead SNP would be the most likely have shared causal variants and not be as subjected to confounding from LD differences between populations. however we do not think it would be interesting to formally test for heterogeneity in this subset given we know *a priori* that there is high correlation of effect sizes and thus this set is biased towards showing no heterogeneity.

3. Fig 2 is very interesting showcasing the main discovery of high correlation. Can the authors also investigate the correlation in the phenotypes (i.e. methylations) across data sets/ancestries? Are they of the same magnitude or lower due to different environments?

We thank the reviewer for their suggestion. We do note that such correlations will be largely driven by methylation sites with very low or very high methylation, and thus we expect these to all be close to one. Using the cohorts available in house we calculated the Pearson's correlation in mean DNAm for each DNAm probes across all cohorts. All cohorts were found to be strongly correlated (correlations ranged from 0.99 to 1). This was performed on DNAm levels prior to adjustment for typical covariates (age, sex, cell type proportions, phenotypes). Given the negligible deviation from 1 for all correlations for the cohorts we had available to analyse inhouse we have not requested analysis for the THCH as we do not think this will provide any further insight. We do not think this will add further value to the manuscript.

Cohort	SGPD	LBC	BSGS
LBC	0.991 ($p < 10^{-100}$)		
BSGS	0.999 ($p < 10^{-100}$)	0.988 ($p < 10^{-100}$)	
CHNMND	0.998 ($p < 10^{-100}$)	0.997 ($p < 10^{-100}$)	0.997 ($p < 10^{-100}$)

4. Discussion about the utility of cross-ancestral data could be enhanced; some of the first studies showing how fine-mapping could be drastically improved in cross ancestry data date back to more than a decade ago in studies such as Zaitlen et al AJHG 2010 (PMID: 20085711).

We have enhanced our discussion of cross-ancestral data in enhancing fine-mapping. The following text has been added:

"The utility of cross-ancestry fine-mapping in reducing the number of putative causal variants has been previously demonstrated for many complex traits [1-7] including schizophrenia [5], type 2 diabetes [8, 9], BMI [10], RA [11] and height [7]. The success of such methods is supported by evidence that that association signals are driven by variants shared across populations [1, 7, 12], despite observed heterogeneity in effect sizes [5, 7, 12]. By utilising DNAm as a model complex trait" ...

Reviewer #2:

Authors conducted methylation QTL (mQTL) study meta-analysis of Europeans and Asians. In total, 5 cohorts with ~5700 individuals were included. Cross-population mQTL meta-analysis identified that mQTL loci were generally shared. Ancestry-specific QTL were mostly due to different allele frequency spectra between populations. They found fine-mapping of putative mQTL by cross-population approaches. Some overlap with the GWAS variants were observed. While this is a typical mQTL study, findings and concepts were not novel.

1. It was not clear for the reviewer what was the main findings and novelty of the study. Certain sample size may provide mQTLs, but how the results overwhelmed the previous similar studies were not clear.

Our study combines multiple cohorts from European and East Asian ancestries with relatively large sample sizes (for mQTL studies). This allows us to investigate shared and non-shared genetic effects across ancestries for increasingly rare genetic variants, which allowed us to demonstrate that differences between ancestral groups was largely due to allele frequencies. Smaller studies can only draw conclusions for common genetic variants and are only powered to investigate SNPs with the largest effect – i.e. the SNPs that may be considered a priori as more likely to show shared effects across ancestries. In this case, bigger is actually better to answer the questions posed in this study.

2. No novel conceptual findings. Cross-population study can provide statistical power. It is not surprising.

While we also were not surprised that cross-population studies improved fine-mapping (which we assume is what the reviewer is referring to in this comment), the quantification of exactly how much improvement was made relative to just increasing sample size is novel – particularly when considering tens of thousands of genetic effects across the whole genome.

3. Authors concluded that majority of the ancestry-specific mQTL were due to allele frequency differences. It was not exciting. More detailed assessments on truly ancestry-specific mQTL with difference effect sizes across populations were warranted.

We thank the reviewer for this comment and applaud their high bar for excitement! We would have loved to perform more detailed assessments of true differences, but our conclusions show that “truly ancestry-specific mQTL with difference effect sizes across populations” are very, very rare. This is an important conclusion, and has implications for the application of polygenic scores across ancestries.

4. In cross-populations comparisons, fine-mapping should be applied.

Our comparison of mQTL across populations did consider multiple levels of shared effects. Particularly, we quantified mQTL that were in overlapping LD blocks. This is effectively fine-mapping in terms of finding shared SNPs across LD blocks. In addition, fine-mapping methods tend to perform poorly in regions with massive genetic effects (e.g. mQTL), and in

meta-analysed data. Hence the simple approach we took when assessing cross-ancestry fine mapping vs sample size increases within ancestries.

5. Functional annotation of the mQTL variants can be done in depth.

While we agree that functionally annotation can be done, it is unclear how such annotation (performed at tens of thousands of mQTL) would add to this study.

6. As for GWAS risk variant overlap, much wider ranges of GWAS phenotypes should be applied.

We thank the reviewer for their suggestion. We have expanded the analysis investigating pleiotropic associations with GWAS risk variants in ancestry specific mQTLs to now include all 220 traits reported in Sakaue et al. (2021) [13]. We note that for some of the traits previously analysed, the GWAS data in individuals of European ancestry now has an expanded sample size due to the meta-analyses of the UK Biobank and FinnGen data.

The results have been edited as follows:

We demonstrate the utility of these ancestry-specific mQTLs by identifying pleiotropic associations with common genetic variation associated with 220 traits from Sakaue et al. (2021) [41] (Supplemental Data 1). Using SMR [42], we tested if the SNP effects on the traits were mediated by DNAm at 28,925 ancestry-specific mQTLs. Across 57 of these traits, we identified 163 ancestry-specific associations with DNAm ($p_{\text{SMR}} < 10^{-6}$), of which 147 were unique in EUR and 16 in EAS with non-significant heterogeneity ($p_{\text{HEIDI}} > 0.01$; Supplementary Data 2). We note that a potential driver of these difference would be low AF in the alternate ancestry which is likely the case for 115 (70.6%) of the associations (108 identified in EUR and 7 in EAS), however the remaining 48 pleiotropic associations occur at SNPs with common AF in both ancestries (minor AF > 0.05). For example, we identified a unique pleiotropic association in individuals of EUR ancestry with probe cg19197236 in chromosome 2 with Glucocorticoid use (R03BA) (Figure 6), with no association for either DNAm or R03BA in individuals of EAS ancestry (minor AF of SNP rs7608524 of 0.37 in EUR and 0.41 in EAS).

The methods have been edited as follows:

To demonstrate the utility of ancestry-specific mQTLs, we investigated the presence of pleiotropic associations between DNAm and 220 traits as reported in Sakaue et al. (2021) [13]. Summary statistics of analysis performed in individuals of EUR ancestry were obtained from meta-analyses with the UK Biobank and FinnGen while EAS was from BioBank Japan (Supplementary Data 1).

7. This study can have a resource value. But no descriptions on data deposit, which should lower the value.

We completely agree with making all research data available. We had included data availability statements in the original manuscript for each of the cohorts but agree we can expand the availability of the outcomes of our study. We have added a data availability statement to the manuscript:

mQTL summary data from the meta-analysis of samples of each European (n=3,071) and East Asian (n=2,099) ancestry are available from <https://yanglab.westlake.edu.cn/software/smr/#mQTLsummarydata>. These results have been provided in SMR BESD format (see <https://yanglab.westlake.edu.cn/software/smr/#BESDformat>). Access to individual level data for each of the cohorts is stipulated in the Methods. The web links for the publicly available datasets used in the study are as follows: mQTL data from Hawe et al 2022 [31]: <https://zenodo.org/record/5196216#.YRZ3TfJxeUk>; mQTL data from Min et al 2021 [81] data: <http://mqtl.db.godmc.org.uk>; GWAS summary statistics for 220 traits used for SMR from Sakaue et al 2021 [41]: <https://pheweb.jp/downloads>; Annotation of Infinium DNA Methylation BeadChip probes: <https://zwdzwd.github.io/InfiniumAnnotation>.

References

1. Peterson, R.E., et al., *Genome-wide Association Studies in Ancestrally Diverse Populations: Opportunities, Methods, Pitfalls, and Recommendations*. Cell, 2019. **179**(3): p. 589-603.
2. Teo, Y.Y., et al., *Identifying candidate causal variants via trans-population fine-mapping*. Genet Epidemiol, 2010. **34**(7): p. 653-64.
3. Zaitlen, N., et al., *Leveraging genetic variability across populations for the identification of causal variants*. Am J Hum Genet, 2010. **86**(1): p. 23-33.
4. Wang, Y.-F., et al., *Identification of 38 novel loci for systemic lupus erythematosus and genetic heterogeneity between ancestral groups*. Nature Communications, 2021. **12**(1): p. 772.
5. Lam, M., et al., *Comparative genetic architectures of schizophrenia in East Asian and European populations*. Nature Genetics, 2019. **51**(12): p. 1670-1678.
6. DIABetes Genetics Replication And Meta-analysis (DIAGRAM) Consortium., et al., *Genome-wide trans-ancestry meta-analysis provides insight into the genetic architecture of type 2 diabetes susceptibility*. Nature Genetics, 2014. **46**(3): p. 234-244.
7. Wojcik, G.L., et al., *Genetic analyses of diverse populations improves discovery for complex traits*. Nature, 2019. **570**(7762): p. 514-518.
8. Twee-Hee Ong, R., et al., *Efficiency of trans-ethnic genome-wide meta-analysis and fine-mapping*. European Journal of Human Genetics, 2012. **20**(12): p. 1300-1307.
9. Mahajan, A., et al., *Genome-wide trans-ancestry meta-analysis provides insight into the genetic architecture of type 2 diabetes susceptibility*. Nat Genet, 2014. **46**(3): p. 234-44.
10. Fernández-Rhodes, L., et al., *Trans-ethnic fine-mapping of genetic loci for body mass index in the diverse ancestral populations of the Population Architecture using Genomics and Epidemiology (PAGE) Study reveals evidence for multiple signals at established loci*. Human Genetics, 2017. **136**(6): p. 771-800.
11. Kichaev, G. and B. Pasaniuc, *Leveraging Functional-Annotation Data in Trans-ethnic Fine-Mapping Studies*. Am J Hum Genet, 2015. **97**(2): p. 260-71.
12. Marigorta, U.M. and A. Navarro, *High trans-ethnic replicability of GWAS results implies common causal variants*. PLoS Genet, 2013. **9**(6): p. e1003566.
13. Sakaue, S., et al., *A cross-population atlas of genetic associations for 220 human phenotypes*. Nature Genetics, 2021. **53**(10): p. 1415-1424.
14. Hawe, J.S., et al., *Genetic variation influencing DNA methylation provides insights into molecular mechanisms regulating genomic function*. Nature Genetics, 2022. **54**(1): p. 18-29.

15. Min, J.L., et al., *Genomic and phenotypic insights from an atlas of genetic effects on DNA methylation*. Nature Genetics, 2021. **53**(9): p. 1311-1321.

REVIEWERS' COMMENTS

Reviewer #1 (Remarks to the Author):

The authors have partially answered my previous comments; the authors have chosen not to pursue my comment 2. Whereas I disagree with their choice re comment2, I maintain that the manuscript is a worth-wile addition to the field and I support its publication.

Reviewer #2 (Remarks to the Author):

Authors well addressed the reviewer's comments